



**Socio-hydrologic data assimilation: Analyzing human-flood interactions by model-**
**data integration**
Yohei Sawada[1], Risa Hanazaki[2]
[1] Institute of Engineering Innovation, School of Engineering, the University of Tokyo,
Tokyo, Japan
[2] Institute of Industrial Science, the University of Tokyo, Tokyo, Japan
Corresponding author: Y. Sawada, Institute of Engineering Innovation, the University of
Tokyo, Tokyo, Japan, 2-11-6, Yayoi, Bunkyo-ku, Tokyo, Japan, yohei.sawada@sogo.t.u-
tokyo.ac.jp





**Abstract**
In socio-hydrology, human-water interactions are simulated by mathematical models.
Although the integration of these socio-hydrologic models and observation data is
necessary to improve the understanding of the human-water interactions, the
methodological development of the model-data integration in socio-hydrology is in its
infancy. Here we propose to apply sequential data assimilation, which has been widely
used in geoscience, to a socio-hydrological model. We developed particle filtering for a
widely adopted flood risk model and performed an idealized observation system
simulation experiment to demonstrate the potential of the sequential data assimilation in
socio-hydrology. In this experiment, the flood risk model's parameters, the input forcing
data, and empirical social data were assumed to be somewhat imperfect. We tested if data
assimilation can contribute to accurately reconstructing the historical human-flood
interactions by integrating these imperfect models and imperfect and sparsely distributed
data. Our results highlight that it is important to sequentially constrain both state variables
and parameters when the input forcing is uncertain. Our proposed method can accurately
estimate the model's unknown parameters even if the true model parameter temporally
varies. The small amount of empirical data can significantly improve the simulation skill



of the flood risk model. Therefore, sequential data assimilation is useful to reconstruct
historical socio-hydrological processes by the synergistic effect of models and data.



## 1. Introduction

Socio-hydrology is an emerging research field in which two-way feedbacks between
social and water systems are investigated (Sivapalan et al. 2012, 2014). Understanding
complex socio-hydrologic phenomena contributes to solving water crises around the
world. Socio-hydrology has been recognized as an important scientific grand challenge
to meet United Nations' Sustainable Development Goals (Di Baldassarre et al. 2019).
The most popular approach in socio-hydrology is to develop dynamic models which
compute non-linear interactions between human and water. For instance, Di Baldassarre
et al. (2013) developed a simplified model, which described human-flood interactions, to
understand the levee effect in which high levees generate a false sense of security and
induce social vulnerabilities to severe floods (see also Viglione et al. 2014; Ciullo et al.
2017). Van Emmerik et al. (2014) developed a stylized model, which described two-way
feedbacks between environment and economic activities, to understand the historical
competition for water between agricultural development and environment health in
Australia (see also Roobavannan et al. 2017). Pande and Savenije (2016) modeled
economic activities of smallholder farmers to analyze the agrarian crisis in Marathwada,
India. While socio-hydrologic models described above assumed the existence of a single





lumped decision maker, Yu et al. (2017) incorporated a collective action into their model
and analyzed the dynamics of community-managed flood protection systems in coastal
Bangladesh. Please refer to Di Baldassarre et al. (2019) for the comprehensive review of
socio-hydrologic modeling.

In addition to these modeling approaches, both qualitative and quantitative data related to
socio-hydrologic processes are important to understand human-water interactions. For
instance, Mostert (2018) revealed historical changes in river management from water
resources development to protection and restoration by analyzing qualitative data. Dang
and Konar (2018) applied econometric methods to analyze quantitative data in both
human and water domains and quantified the causal relationship between trade openness
and water use. Kreibich et al. (2017) performed the detailed case study analysis on paired
floods, consecutive flood events which occurred in the same region with the second flood
causing significantly lower damage. They found that the reduction of vulnerability played
a key role for successful adaptation to the second floods.

Although it is expected that the integration of model and data contributes to accurately
understanding the socio-hydrologic processes (Mount et al. 2016), the methodological



development of the model-data integration in socio-hydrology is in its infancy. Generally,
mathematical models can provide spatiotemporally continuous state variables and
quantitative scenarios for future socio-hydrologic developments. In addition,
mathematical models can quantitatively provide possible scenarios unrealized in the real-
world, which gives the insight to targeted processes (e.g., Viglione et al. 2014). The major
limitation of socio-hydrological models is that they are often inaccurate due to the
uncertainty in their input forcing, parameters, and descriptions of the processes. On the
other hand, hydrologic and social data are often more reliable than numerical models and
can provide more complete understanding of the socio-hydrological processes (e.g.,
Mostert 2018), although data also have uncertainties. However, in many cases, relevant
data in socio-hydrology are sparsely distributed so that it is difficult to completely
reconstruct the historical socio-hydrologic processes from data. The other limitation of
the data-driven approach is that the quantification of the causal relationship cannot be
easily done only by empirical data (e.g., Dang and Konar 2018). Considering this
advantages and disadvantages of model and data, previous studies used social statistics
to calibrate and validate their socio-hydrologic models (e.g., Barendrecht et al. 2019;
Roobavannan et al. 2017; Ciullo et al. 2017; van Emmerik et al. 2014; Gonzales and
Ajami 2017).




In geosciences, sequential data assimilation has been widely used for the model-data
integration. Data assimilation sequentially adjusts the predicted state variables and
parameters of dynamic models by integrating observation data into models based on
Bayes' theorem. Data assimilation has been widely applied to numerical weather
prediction (e.g., Miyoshi and Yamane 2007; Bauer et al. 2015; Poterjoy et al. 2019;
Sawada et al. 2019), atmospheric reanalysis (e.g., Kobayashi et al. 2015; Hersbach et al.
2019), and hydrology and land surface modeling (e.g., Moradkhani et al. 2005; Sawada
et al. 2015; Rasmussen et al. 2015; Lievens et al. 2017). Applicability of the data
assimilation approach to the socio-hydrologic models has yet to be investigated.

In this study, we aim to develop the methodology of sequential data assimilation for the
flood risk model proposed by Di Baldassarre et al. (2013). From a series of idealized
experiments, we demonstrate the potential of data assimilation to accurately reconstruct
the historical human-flood interactions. We focus on the case in which the socio-
hydrologic model's parameters, input forcing data, and social data are somewhat
inaccurate.

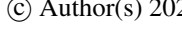


## 2. Method

### 2.1. Model

In this study, we used a socio-hydrologic flood risk model proposed by Di Baldassarre et
al. (2013). This model conceptualizes human-flood interactions by the set of simple
equations which describe the states of flood, economy, technology, politics, and society.
Based on this original model of Di Baldassarre et al. (2013), many similar flood risk
models have been proposed, validated, and applied (e.g., Viglione et al. 2014; Ciullo et
al. 2017; Barendrecht et al. 2019). Here we briefly describe this model. Please refer to Di
Baldassarre et al. (2013) for the complete description of this model.

The governing equations of the flood risk model are shown below:

$$F = \begin{cases} 1 - \exp\left(-\frac{W + \xi_H H}{\alpha_H D}\right) & if\ W + \xi_H H > H \\ 0 & if\ W + \xi_H H \leq H \end{cases} \qquad (1)$$


$$R = \begin{cases} \varepsilon_T (W + \xi_H H - H) & if\ (F > 0)\ and\ (FG > \gamma_E R\sqrt{G})\ and\ (G - FG > \gamma_E R\sqrt{G}) \\ 0 & otherwise \end{cases}$$

122     (2)


$$S = \begin{cases} \alpha_S F & if\ (R > 0) \\ F & if\ (R = 0) \end{cases} \qquad (3)$$

$\frac{dG}{dt} = \rho_E \left(1 - \frac{D}{\lambda_E}\right) G - \Delta(\Upsilon(t))(FG + \gamma_E R\sqrt{G}) \qquad (4)$
$$\frac{dD}{dt} = \left(M - \frac{D}{\lambda_P}\right)\frac{\varphi_P}{\sqrt{G}}$$ (5)
$$\frac{dH}{dt} = \Delta\big(\Upsilon(t)\big)R - \kappa_T H$$ (6)
$$\frac{dM}{dt} = \Delta\big(\Upsilon(t)\big)S - \mu_S M$$ (7)

This model has four state variables: G, D, H, and M. $G(t)$ $[L^2]$ is the size of the human
settlement; $D(t)$ $[L]$ is the distance of the center of mass of the human settlement from the
river; $H(t)$ $[L]$ is the flood protection level (or levee height); $M(t)$ $[.]$ is the social
awareness of the flood risk.

Equation (1) calculates the intensity of flooding events $F(t)$ $[.]$ from the high water level
$W(t)$ $[L]$, the height of the levee $H(t)$ $[L]$, and the distance of the human settlement from
the river $D(t)$ $[L]$. Equation (2) calculates $R(t)$ $[L]$, the amount by which the levees are
raised responding to the flood event. There are three required conditions under which
people decide to raise the levee. First, the flood event occurs. Second, the damage of flood
(FG) should be larger than the cost of raising levee. Third, the cost of raising levee should
be lower than the wealth remaining after the flooding. Equation (3) shows the magnitude
of the psychological shock by the flood event $S(t)$ $[.]$. If the levee is raised, the
psychological shock is assumed to be mitigated. Equation (4) explains the dynamics of



G(t), the size of the human settlement or the wealth of the community. Following the
notation of Di Baldassarre et al. (2013), $\Delta(\Upsilon(t)) = 1$ with integral only when time t
passes the time of the flooding event (F>0), otherwise, $\Delta(\Upsilon(t)) = 0$. The term $FG +$
$\gamma_E R\sqrt{G}$ (total cost of flood damage and construction of levees) appears only if flood
occurs. Equation (5) shows the dynamics of the distance of the center of mass of the
human settlement from the river D(t). When the social awareness of the flood risk is high,
people tend to live far from the river. Equation (6) computes the dynamics of the flood
protection level H(t) and equation (7) shows the dynamics of the social awareness of the
flood risk M(t). The explanation of parameters can be found in Table 1.


**2.2. Data Assimilation**
In this study, we used Sampling Importance Resampling Particle Filtering (SIRPF) as the
method of data assimilation. SIRPF has been widely used in hydrologic data assimilation
(e.g., Moradkhani et al. 2005; Qin et al. 2009; Sawada et al. 2015). Compared with the
other data assimilation algorithms such as ensemble Kalman filter, SIRPF is robust
against model nonlinearity and associated non-Gaussian error distribution. The
disadvantage of SIRPF is that the infeasible computational resources are required if the



numerical model is computationally expensive, which is not the case in the flood risk
model.

The flood risk model can be formulated as a discrete state-space dynamic system:
$x(t + 1) = f\big(x(t), \theta, u(t)\big) + q(t)$                (8)
where $x(t)$ is the state variables (i.e. G, D, H, and M), $\theta$ is the model parameters, $u(t)$
is the external forcing (i.e., the high water level), and $q(t)$ is the noise process which
represents the model error. In data assimilation, it is useful to formulate an observation
process as follows:
$y^f(t) = h\big(x(t)\big) + r(t)$                (9)
where $y^f(t)$ is the simulated observation, h is the observation operator which maps the
model's state variables into the observable variables, and $r(t)$ is the noise process which
represents the observation error.

The SIRPF is a Monte Carlo approximation of Bayesian update of the state variables and
parameters:
$p(x(t), \theta | y^o(1:t)) \propto p(y^o(t) | x(t), \theta) p(x(t), \theta | y^o(1:t-1))$        (10)





where $p(\boldsymbol{x}(t), \boldsymbol{\theta}|\boldsymbol{y}^o(1:t))$ is the posterior probability of the state variables $\boldsymbol{x}(t)$ and
parameters $\boldsymbol{\theta}$ given all observations up to time t $\boldsymbol{y}^o(1:t)$. The prior knowledge,
$p(\boldsymbol{x}(t), \boldsymbol{\theta}|\boldsymbol{y}^o(1:t-1))$, based on the model integration is updated using the likelihood
which includes the new observation at time t $p(\boldsymbol{y}^o(t)|\boldsymbol{x}(t), \boldsymbol{\theta})$. In this study, we assumed
that our observation error follows Gaussian distribution so that the likelihood can be
formulated as follows:
$p(\boldsymbol{y}^o(t)|\boldsymbol{x}(t), \boldsymbol{\theta}) \equiv L(\boldsymbol{y}^o(t), \boldsymbol{x}(t), \boldsymbol{\theta}) =$
$\frac{1}{\sqrt{\det(2\pi\boldsymbol{R})}} \exp\left[-\frac{1}{2}\left(\boldsymbol{y}^o(t) - \boldsymbol{y}^f(t)\right)^T \boldsymbol{R}^{-1}\left(\boldsymbol{y}^o(t) - \boldsymbol{y}^f(t)\right)\right]$ (11)
where **R** is the covariance matrix of the observation error process $\boldsymbol{r}(t)$. The prior
knowledge of the state variables is approximated by the ensemble simulation:
$p(\boldsymbol{x}(t)|\boldsymbol{y}^o(1:t-1)) \approx \frac{1}{N}\sum_{i=1}^{N} \delta\left[\boldsymbol{x}(t) - f\left(\boldsymbol{x}^i(t-1), \boldsymbol{\theta}^i, \boldsymbol{u}^i(t-1)\right)\right]$     (12)
where N is the ensemble size, $\boldsymbol{x}^i, \boldsymbol{\theta}^i, \boldsymbol{u}^i$ are the realizations of the ensemble member i,
and $\delta[.]$ is the Direc delta function.

The posterior probability of the state variables and parameters can be approximated as
follows:
$p(\boldsymbol{x}(t)|\boldsymbol{y}^o(1:t)) \approx \sum_{i=1}^{N} w(i)\delta(\boldsymbol{x}(t) - \boldsymbol{x}^i(t))$     (13)
$p(\boldsymbol{\theta}|\boldsymbol{y}^o(1:t)) \approx \sum_{i=1}^{N} w(i)\delta(\boldsymbol{\theta} - \boldsymbol{\theta}^i)$     (14)





where $w(i)$ is the normalized weight for the realization of the ensemble member i and
is calculated using the likelihood (see also equation (11)).
$$w(i) = \frac{L(\boldsymbol{y}^o(t), \boldsymbol{x}^i(t), \theta^i)}{\sum_{k=1}^{N} L(\boldsymbol{y}^o(t), \boldsymbol{x}^k(t), \theta^k)} \qquad (15)$$

The implementation of SIRPF is the following:

1.  Model state variables are updated from time t-1 to t using ensemble

simulation (equations (8) and (12)).

2.  Simulated observations are calculated for all ensembles (equation (9)).

3.  The likelihood for each ensemble member is calculated (equation (11))

4.  The weights are obtained for all ensembles (equation (15))

5.  We applied a resampling procedure according to the normalized weights.

The normalized weights of ensemble i, $w(i)$, can be recognized as the

probability that the ensemble i is selected after resampling. Resampled state

variables and parameters are defined as $\boldsymbol{x}^i_{resamp}$ and $\boldsymbol{\theta}^i_{resamp}$, respectively.

6.  Since there are no mechanisms to increase the variance of parameters of

ensemble members, Moradkhani et al. (2005) proposed to perturb the

ensembles of parameters:

$$\boldsymbol{\theta}^i \leftarrow \boldsymbol{\theta}^i_{resamp} + \varepsilon^i \qquad (16)$$

$$\varepsilon^i \sim N(0, \max(\boldsymbol{\omega}, s \times Var^\theta)) \tag{17}$$
where $N(.)$ is the Gaussian distribution, $Var^\theta$ is the variance of $\boldsymbol{\theta}^i$, $\boldsymbol{\omega}$
is the fixed hyperparameter (see Table 1 for its variable) which guarantees
that the ensembles of parameters do not converge into a single value. $s$ is
an adaptively changed factor according to the effective ensemble size, $N_{eff}$.
$$s = s_0\left(1 - \left(\frac{N_{eff}}{N}\right)^2\right) \tag{18}$$
$$N_{eff} = \frac{1}{\sum_{i=1}^{N} w(i)} \tag{19}$$
where $s_0 = 0.05$. The effective ensemble size is the measure of the
diversity of ensembles. If the effective ensemble size becomes small,
ensembles should be strongly perturbed in order to maintain the diversity of
ensembles. Similar strategy has been used in many SIRPF systems (e.g.,
Moradkhani et al. 2005; Poterjoy et al. 2019).


**3. Experiment design**
In this study, we performed three observation system simulation experiments (OSSEs).
In the OSSE, we generated the synthetic truth of the state and flux variables by driving
the flood risk model with the specified parameters and input. Then, we generated





synthetic observations by adding the noise to this synthetic truth. Those synthetic
observations were assimilated into the model by SIRPF. The performance of SIRPF was
evaluated by comparing the estimated state variables by SIRPF with the synthetic truth.
Model parameters used to generate the synthetic truth can be found in Table 1. They are
identical to Di Baldassarre et al. (2013). The OSSE has been recognized as an important
preliminary step to verify the newly developed data assimilation systems (e.g.,
Moradkhani et al. 2005; Vrugt et al. 2013; Penny and Miyoshi 2016; Sawada et al. 2018).

The high water level for the synthetic truth was generated by the following:
$W = \min (v - 10, 0)$                  (20)
$v$  follows the Gumbel distribution:
$p(v) = \frac{\exp\left(-\frac{v-\mu}{\beta}\right)}{\beta} \exp\left(-\exp(-(v - \mu)\beta)\right)$       (21)
where  $\mu = 9, \beta = 2.5$. Although our high water level is not identical to Di Baldassarre
et al. (2013), the estimated trajectory of the state variables is similar to Di Baldassarre et
al. (2013).

Synthetic observations were generated by adding the Gaussian white noise to the F, G, D,
H, and M (see section 2.1) of the synthetic truth. The mean of the Gaussian white noise



was 0. The variance of the Gaussian white noise was 10% of the synthetic true variables.
We firstly assumed that all of the F, G, D, H, and M can be observed every 10 years or
every 10 model integration steps. Then, we evaluated the sensitivity of the observation
network (i.e. the observable variables and the observation intervals) to the SIRPF's
performance.

We used the ensemble mean of root-mean square errors (mRMSE) as an evaluation
metrics:
$RMSE^i = \sqrt{\frac{1}{T}\sum_{t=1}^{T}(x^i(t) - z(t))}$           (22)
$mRMSE = \frac{1}{N}\sum_{i=1}^{N} RMSE^i$           (23)
where $RMSE^i$ is root-mean-square-error for i th ensemble, T is the computational period,
$x^i(t)$ is the simulated state variables of ensemble i at time t, $z(t)$ is the synthetic truth
at time t.




**3.1. Experiment 1: Perfect model with uncertain high water levels**



In the first OSSE, we assumed that the model was perfect, and we knew it. We used the
same parameter variables as the synthetic truth run and we did not perform the estimation
of parameters. Our SIRPF estimated only state variables. Although the model had no
uncertainty, it was assumed that the input data, the timeseries of the high water level, were
uncertain. Lognormal multiplicative noise was added to the synthetic true high water level
so that different ensemble members have different high water levels in the data
assimilation experiment. The two parameters of the lognormal distribution, commonly
called $\mu$ and $\sigma$, were set to 0 and 0.15, respectively.


**3.2. Experiment 2: Unknown model parameters and uncertain high water levels**
In the second OSSE, we assumed that some of the synthetic true parameter values were
unknown. The unknown parameters in the experiment 2 were the cost of levee raising $\gamma_E$,
the rate by which new properties can be built $\varphi_P$, the rate of decay of levees $\kappa_T$, and
memory loss rate $\mu_S$ (see Table 1). We selected these unknown parameters one by one
from four equations of economy, politics, technology, and social (see section 2.1). The
initial parameter variables were assumed to be distributed in the bounded uniform
distributions whose ranges were found in Table 1. Our SIRPF sequentially assimilated
observations and estimated both state variables and parameters in the experiment 2. The
high water level data were uncertain as the experiment 1.


**3.3. Experiment 3: Unknown and time-variant model parameters and uncertain**
**high water levels**
To further demonstrate the potential of sequential data assimilation in socio-hydrology,
we assumed that the description of the model was biased in the experiment 3. Here we
assumed that one of the model parameters was temporally varied by the unknown
dynamics. Specifically, the memory loss rate, $\mu_S$, was temporally varied in the
experiment 3:
$$\mu_S(t) = \begin{cases} 0.01 \ (t < 250) \\ 0.01 + (t - 250) \times \frac{0.10 - 0.01}{500} \ (250 \leq t < 750) \\ 0.10 \ (750 \leq t) \end{cases} \quad (24)$$
In this problem setting, we misunderstood the memory loss rate as a time-invariant
parameter in our socio-hydrological model since the dynamics to control the memory loss
rate was unknown. We evaluated if SIRPF could track this time-variant parameter and
reveal the bias of the model's description. The cost of levee raising $\gamma_E$, the rate by which
new properties can be built $\varphi_P$, and the rate of decay of levees $\kappa_T$ were assumed to be





time-invariant unknown parameters as they were in the experiment 2. The input forcing
data, high water level, were uncertain as described in the experiment 1.


**4. Results**
**4.1. Experiment 1: Perfect model with uncertain high water levels**
Figure 1 shows the timeseries of the model variables calculated by 5000 ensembles with
no data assimilation. Although the ensemble mean of the state variables is close to the
synthetic truth, the ensembles have the large spread especially for G. The uncertainty in
the input forcing brings the uncertainty in the estimation of the historical socio-hydrologic
condition.

Figure 2 indicates that this uncertainty is mitigated by assimilating the observations of F,
G, D, H, and M into the model every 10 years with 5000 ensembles. Table 2 shows that
RMSE is reduced for all state variables by data assimilation.

While we can observe all of F, G, D, H, and M in Figure 2 and Table 2, Figure 3 shows
the performance of our SIRPF in which only one of them can be observed. Figure 3





reveals that we can accurately propagate the observation information into the model state
space. In other words, our SIRPF can positively impact the estimation of not only
observed state variables but unobserved state variables. For instance, even if we can
observe only G, the simulation of all G, D, H, and M is improved. This finding is
promising since all of the state variables cannot be observed in the real-world applications.
Figure 3 also shows that observing F is not effective compared with the other variables.
This is because F is a flux and F can be observed only when floods occur so that the
number of effective observations is small. In addition, H is decoupled from the other state
variables. Observing F, D, and M negatively impacts the estimation of H and observing
H does not significantly improve the simulation of D and M. This is because the dynamics
of H is largely determined by high water levels whose uncertainty is not mitigated by our
SIRPF system.

While we can observe every 10 years in Figure 2 and Table 2, Figure 4 shows the
sensitivity of the observation intervals to the performance of our SIRPF. Our SIRPF
improves the estimation of the state variables when we can obtain observation once in
50-year or 100-year (see also Figure S1 for timeseries of the model's variables), which is
promising since we cannot expect the frequent observations in the real-world applications.




Although we demonstrate the potential of our SIRPF with 5000 ensembles thus far, the
improvement of the simulation skill can be found in much smaller ensemble sizes. The
performance of our SIRPF with 20 ensembles is similar to that with 5000 ensembles
(Figure S2).


**4.2. Experiment 2: Unknown model parameters and uncertain high water levels**
Figure 5 reveals that the flood risk model completely loses its skill to estimate the human-
flood interactions if there are uncertainties in model parameters and high water levels
prescribed in Section 3. In contrast to the experiment 1, the ensemble mean cannot
accurately reproduce the synthetic truth.

Figure 6 indicates that our SIRPF can accurately estimate the model state variables by
assimilating the observations of F, G, D, H, and M into the model every 10 years with
5000 ensembles. Figure 7 indicates that four unknown parameters can also be accurately
estimated. We find that it is relatively difficult to estimate the rate of levee's decay, $\kappa_T$,
compared with the other parameters. This is because $\kappa_T$ strongly affects the dynamics





of H and the uncertainty in H is largely determined by the uncertainty in high water levels,
which is not directly mitigated by our SIRPF system. Table 3 shows that RMSE is reduced
for both state variables and parameters by data assimilation.

We analyzed the impacts of the individual observation types on the simulation skill as we
did in the experiment 1. Figure 8a shows that the effects of the individual observation
types are similar to what we found in the experiment 1: (1) our SIRPF can improve the
skill to simulate unobservable state variables; (2) observing F is not effective compared
with the other observations; (3) H is decoupled from the other state variables. Figure 8b
reveals that the parameters can be efficiently estimated by assimilating the observation of
the state variables which are tightly related to the targeted parameters. For instance,
observing D can greatly improve the rate by which new properties can be built, $\varphi_P$, in
equation (5) which governs the dynamics of D. However, assimilating a single
observation type can contribute to accurately estimating all four parameters in many cases,
which is the promising result considering the sparsity of the observation in the real-world
applications.



The good performance of our SIRPF can be found with the longer observation intervals
as we found in the experiment 1. Figure 9 indicates that our SIRPF can improve the
estimation of the state variables and parameters when we can obtain observation once in
50-year or 100-year (see also Figures S3 and S4 for timeseries of the model's variables).

In contrast to the experiment 1, the larger ensemble size is required to stably estimate both
state variables and parameters (Figure S5). The increased degree of freedom and the
nonlinear relationship between parameters and observations increase the necessary
ensemble size.


**4.3. Experiment 3: Unknown and time-variant model parameters and uncertain**
**high water levels**
In addition to the experiment 2, one of the unknown parameters ($\mu_S$) temporally varies in
the synthetic truth of the experiment 3. Figure 10 and Table 4 indicate that despite the
error in the model's description, our SIRPF can greatly improve the simulation of the
flood risk model. Please note that the synthetic truth shown in Figure 10 is different from
that of the previous experiments especially for D and M. Figure 11d indicates that we can





accurately estimate the time-variant parameter ($\mu_S$) as well as the other time-invariant
parameters (Figures 11a, 11b, and 11c). This result is promising since we cannot expect
the perfect description of the socio-hydrologic model in the real-world applications. We
also performed the sensitivity test on observation types, observation intervals, and
ensemble sizes, which results in the same conclusions as the experiment 2 (not shown).


**5. Discussion**
In this study, we developed the sequential data assimilation system for the widely adopted
socio-hydrological model, the flood risk model by Di Baldassarre et al. (2013). We
demonstrated that our SIRPF for the flood risk model is useful to reconstruct the historical
human-flood interactions, which can be called "socio-hydrologic reanalysis", by
integrating sparsely distributed observations and imperfect numerical simulation.
Although our experiment design was idealized, this study reveals several important
findings toward real-world applications.

First, the sequential data assimilation can mitigate the negative impact of the uncertainty
in the input forcing on the simulation of socio-hydrologic state variables. We found that



the small perturbation of high water levels greatly affects the long-term trajectory of the
socio-hydrologic state variables as Viglione et al. (2014) found. It is necessary to
sequentially constrain the state variables and parameters by sequential data assimilation
if the input forcing is uncertain although previous studies on the model-data integration
in socio-hydrology mainly focused on parameter calibration assuming no uncertainty in
the input forcing (e.g., Barendrecht et al. 2019; Roobavannan et al. 2017; Ciullo et al.
2017; van Emmerik et al. 2014; Gonzales and Ajami 2017). To deeply understand the
socio-hydrologic processes, the long-term historical analysis should be performed.
Although there are many studies on the accurate reconstruction of the historical weather
condition (e.g., Toride et al. 2017), it may be necessary to tackle with the uncertainty in
hydrometeorological datasets used for the input forcing of the socio-hydrologic models.

Second, our SIRPF can efficiently improve the simulation of the socio-hydrologic state
variables using the sparsely distributed data. All model variables should not necessarily
be observed to constrain the model's state variables and parameters. In some cases,
observations of a single state variable are enough to reconstruct the accurate socio-
hydrologic state. In addition, observation intervals can be longer than 10-year. Since it is
difficult to obtain the large volume of data in socio-hydrology, this finding is promising



toward real-world applications. We also give some insights about the informative
observation types in the flood risk model. With uncertain high water levels, observations
of the intensity of flooding events F and the height of levee H are not informative (i.e. the
assimilation of these observations cannot greatly improve the simulation skill) although
the empirical data which can be related to F and H may be easily found. On the other
hand, observations of the size of the human settlement G are informative to constrain the
flood risk model. Model parameters can be efficiently estimated by assimilating the state
variables which is tightly related to the targeted parameters, which is consistent to the
findings of the idealized experiment by Barendrecht et al. (2019).

Third, our SIRPF is robust to the imperfectness of the socio-hydrologic model. The
unknown parameters can be efficiently estimated by the sequential data assimilation.
While previous studies evaluated the trajectory in the whole study period to calibrate the
socio-hydrologic models by iteratively performing the long-term model integration (e.g.,
Barendrecht et al. 2019; Roobavannan et al. 2017; Ciullo et al. 2017; van Emmerik et al.
2014; Gonzales and Ajami 2017), we sequentially optimize parameters based on the
relatively short-term timeseries allowing parameters to temporally vary in the study
period. The advantage of this strategy is that we can deal with time-variant parameters as



previously demonstrated in the applications to hydrologic models (e.g., Pathiraja et al.
2018). In the model development, parameters are formulated as time-invariant values so
that the existence of time-variant parameters indicates the imperfect description of
dynamic models. Sequential data assimilation can mitigate the negative impact of this
imperfect model description. Vrugt et al. (2013) pointed out that the parameter
optimization by the sequential filters is unstable if parameter sensitivity temporally
changes (e.g., parameters affects the model's dynamics differently in the different
seasons), which may be the potential limitation of our strategy compared with Bayesian
inference based on the long-term trajectory such as Barendrecht et al. (2019).


**6. Conclusion**
In this study, we proposed to apply the sequential data assimilation to the socio-
hydrologic models. By several OSSEs in the flood risk modeling, we found that our
proposed SIRPF is robust to the imperfect input forcing and the imperfect model. The
sequential data assimilation is useful to reconstruct the socio-hydrologic conditions from
the inaccurate and sparsely distributed data and the imperfect simulation. Future work
will focus on the verification of our approach by the real data.





**Acknowledgements**
We thank Di Baldassarre for sharing the original source code of the flood risk model.
Data Integration and Analysis System (DIAS) provided us the computational resources.

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

2017









**Table 1.** Parameters of the flood risk model

|  | description | Values | Ranges in data assimilation | $\omega$ in equation (17) |
|---|---|---|---|---|
| $\xi_H$ | proportion of additional high water level due to levee heightening | 0.5 | - | - |
| $\alpha_H$ | parameter related to the slope of the floodplain and the resilience of the human settlement | 0.01 | - | - |
| $\rho_E$ | maximum relative growth rate | 0.02 | - | - |
| $\lambda_E$ | critical distance from the river beyond which the settlement can no longer grow | 5000 | - | - |
| $\gamma_E$ | Cost of levee raising | 0.5 | 0.2-5.0 | 0.01 |
| $\lambda_P$ | distance at which people would accept to live when they remember past floods whose total consequences were perceived as a total destruction of the settlement | 12000 | - | |
| $\varphi_P$ | rate by which new properties can be built | 10000 | 1000-50000 | 100 |
| $\varepsilon_T$ | safety factor for levees rising | 1.1 | - | - |
| $\kappa_T$ | rate of decay of levees | 0.001 | 0-0.0015 | 0.0000025 |
| $\alpha_S$ | proportion of shock after flooding if levees are risen | 0.5 | - | - |
| $\mu_S$ | memory loss rate | 0.05 | 0-0.4 | 0.0025 |







**Table 2.** RMSE of the no data assimilation experiment (NoDA) and the data
assimilation experiment (DA) in which all observations are assimilated every 10 years
with 5000 ensembles in the experiment 1 (see section 3.1).

|   | NoDA | DA |
|---|---|---|
| G | $1.06 \times 10^6$ | $1.64 \times 10^4$ |
| D | $3.60 \times 10^2$ | $3.92 \times 10^1$ |
| H | 2.65 | 1.41 |
| M | $1.08 \times 10^{-1}$ | $8.32 \times 10^{-2}$ |






**Table 3.** RMSE of the no data assimilation experiment (NoDA) and the data
assimilation experiment (DA) in which all observations are assimilated every 10 years
with 5000 ensembles in the experiment 2 (see section 3.2).

|            | NoDA                  | DA                     |
|------------|-----------------------|------------------------|
| G          | $2.97 \times 10^6$    | $1.64 \times 10^4$     |
| D          | $1.86 \times 10^3$    | $1.01 \times 10^2$     |
| H          | 9.35                  | 1.63                   |
| M          | $2.24 \times 10^{-1}$ | $8.99 \times 10^{-2}$  |
| $\gamma_E$ | 2.08                  | $4.27 \times 10^{-1}$  |
| $\varphi_P$| $1.72 \times 10^4$    | $3.81 \times 10^3$     |
| $\kappa_T$ | $4.12 \times 10^{-4}$ | $2.36 \times 10^{-4}$  |
| $\mu_S$    | $1.55 \times 10^{-1}$ | $2.43 \times 10^{-2}$  |






**Table 4.** RMSE of the no data assimilation experiment (NoDA) and the data
assimilation experiment (DA) in which all observations are assimilated every 10 years
with 5000 ensembles in the experiment 3 (see section 3.3).

|  | NoDA | DA |
|---|---|---|
| G | $2.90 \times 10^6$ | $3.78 \times 10^3$ |
| D | $2.12 \times 10^3$ | $1.45 \times 10^2$ |
| H | 9.33 | 1.62 |
| M | $2.45 \times 10^{-1}$ | $7.70 \times 10^{-2}$ |
| $\gamma_E$ | 2.08 | $4.51 \times 10^{-1}$ |
| $\varphi_P$ | $1.72 \times 10^4$ | $5.00 \times 10^3$ |
| $\kappa_T$ | $4.12 \times 10^{-4}$ | $2.77 \times 10^{-4}$ |
| $\mu_S$ | $1.60 \times 10^{-1}$ | $3.22 \times 10^{-2}$ |


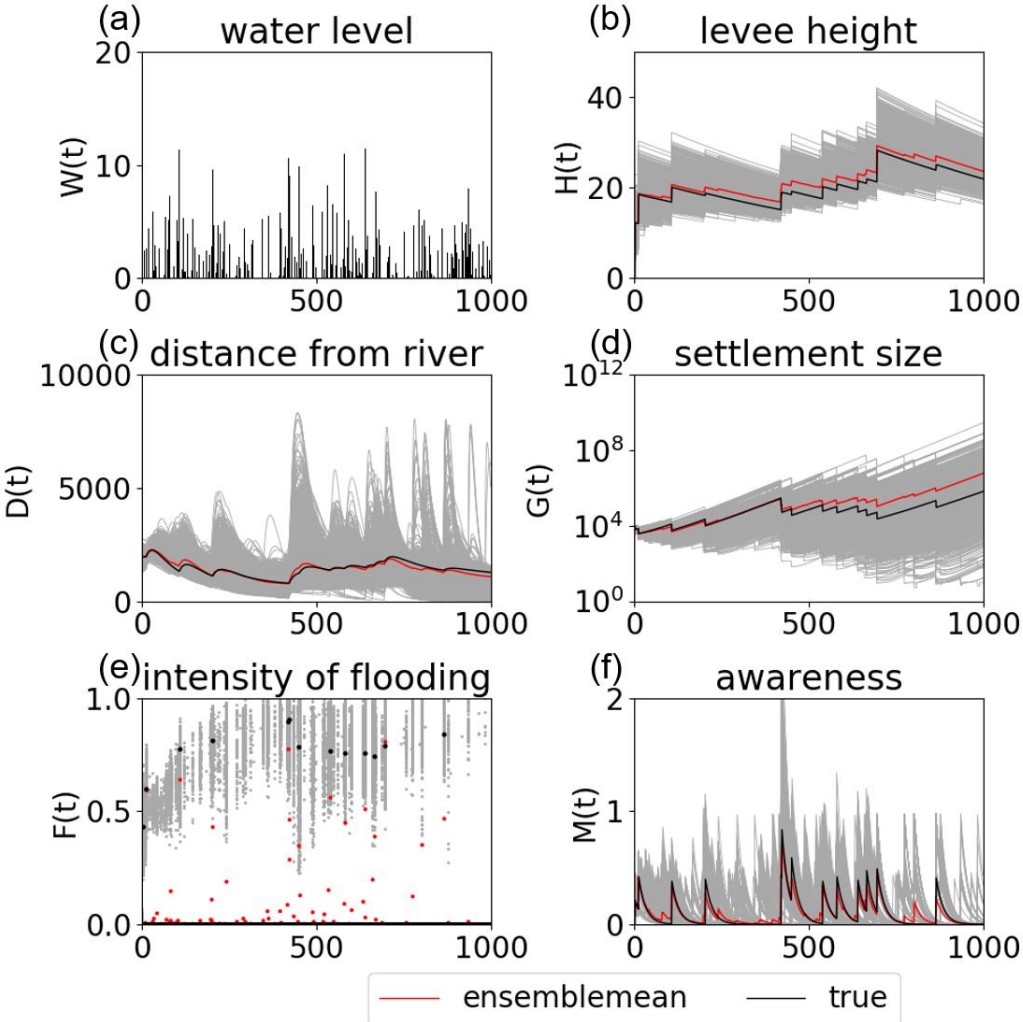


**Figure 1**. Timeseries of (a) high water level W(t), (b) the flood protection level (or levee height) H(t), (c) the

distance of the center of mass of the human settlement from the river D(t), (d) the size of the human settlement

G(t), (e) the intensity of flooding events F(t), and (f) the social awareness of the flood risk M(t) simulated by

5000 ensembles with uncertain high water levels and no data assimilation in the experiment 1 (see section

3.1). Grey, red, and black lines are the ensemble members, their mean, and the synthetic truth, respectively.





**Figure 2**. Timeseries of (a) high water level W(t), (b) the flood protection level (or levee height) H(t), (c) the

distance of the center of mass of the human settlement from the river D(t), (d) the size of the human settlement

G(t), (e) the intensity of flooding events F(t), and (f) the social awareness of the flood risk M(t) simulated by

the data assimilation experiment in which the observations of F, G, D, H, and M are assimilated into the model





every 10 years with 5000 ensembles in the experiment 1 (see section 3.1).Grey, red, and black lines are the
ensemble members, their mean, and the synthetic truth, respectively.





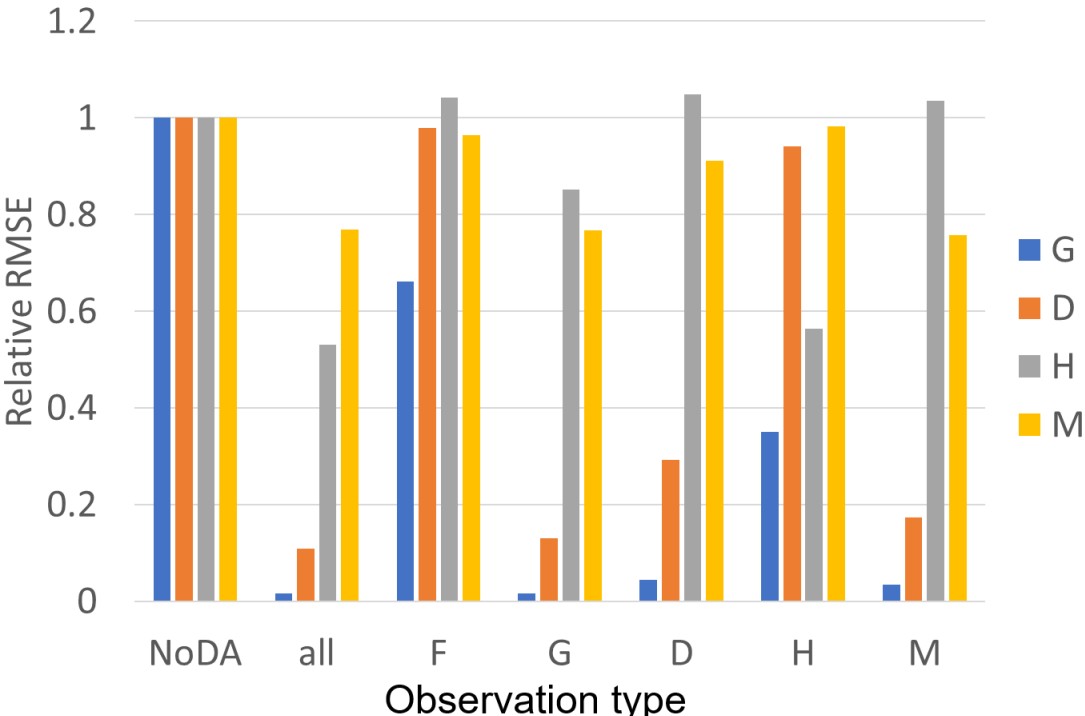


**Figure 3.** The ratio of RMSEs of the no data assimilation experiment (NoDA) to those of the data assimilation

experiments in which all of observations (F, G, D, H, and M) are assimilated (all) and each one of them is
assimilated in the experiment 1 (see section 3.1). Blue, orange, gray, and yellow bars are RMSEs of the size
of the human settlement G(t), the center of mass of the human settlement from the river D(t), the flood
protection level (or levee height) H(t), and the social awareness of the flood risk M(t).



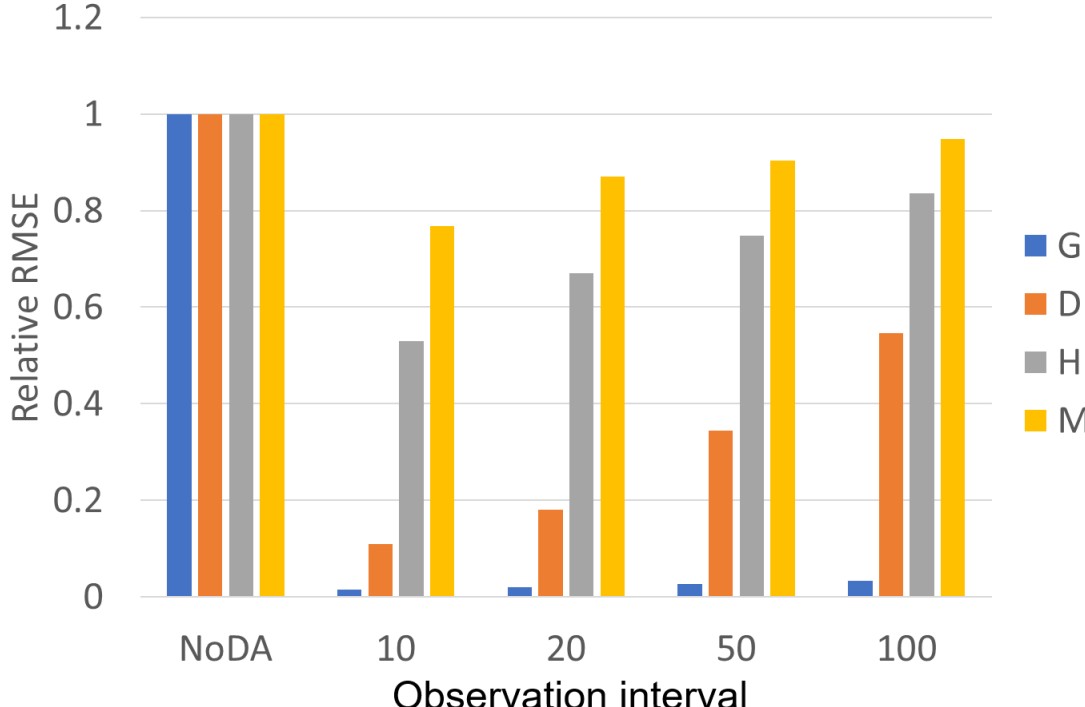


**Figure 4.** The ratio of RMSEs of the no data assimilation experiment (NoDA) to those of the data assimilation

experiments in which all of observations (F, G, D, H, and M) are assimilated every 10, 20, 50, and 100 years

in the experiment 1 (see section 3.1). Blue, orange, gray, and yellow bars are RMSEs of the size of the human

settlement $G(t)$, the center of mass of the human settlement from the river $D(t)$, the flood protection level (or

levee height) $H(t)$, and the social awareness of the flood risk $M(t)$.



**Figure 5.** Timeseries of (a) high water level W(t), (b) the flood protection level (or levee height) H(t), (c) the

distance of the center of mass of the human settlement from the river D(t), (d) the size of the human settlement

G(t), (e) the intensity of flooding events F(t), and (f) the social awareness of the flood risk M(t) simulated by

5000 ensembles with uncertain high water levels and no data assimilation in the experiment 2 (see section

3.2). Grey, red, and black lines are the ensemble members, their mean, and the synthetic truth, respectively.



**Figure 6.** Timeseries of (a) high water level W(t), (b) the flood protection level (or levee height) H(t), (c) the

distance of the center of mass of the human settlement from the river D(t), (d) the size of the human settlement

G(t), (e) the intensity of flooding events F(t), and (f) the social awareness of the flood risk M(t) simulated by

the data assimilation experiment in which the observations of F, G, D, H, and M are assimilated into the model





every 10 years with 5000 ensembles in the experiment 2 (see section 3.2). Grey, red, and black lines are the
ensemble members, their mean, and the synthetic truth, respectively.





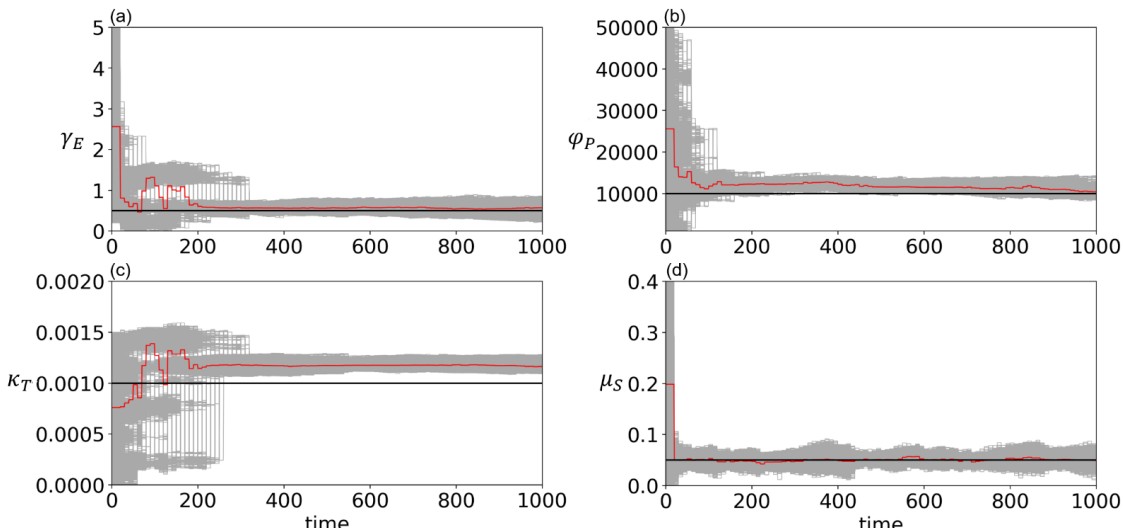


**Figure 7**. Timeseries of (a) the cost of levee raising $\gamma_E$, (b) the rate by which new properties can be built $\varphi_P$,

(c) the rate of decay of levees $\kappa_T$, (d) memory loss rate $\mu_S$ estimated by the data assimilation of all
observations (F, G, D, H, and M) with 5000 ensembles every 10 years in the experiment 2 (see section 3.2).
Grey, red, and black lines are the ensemble members, their mean, and the synthetic truth, respectively.




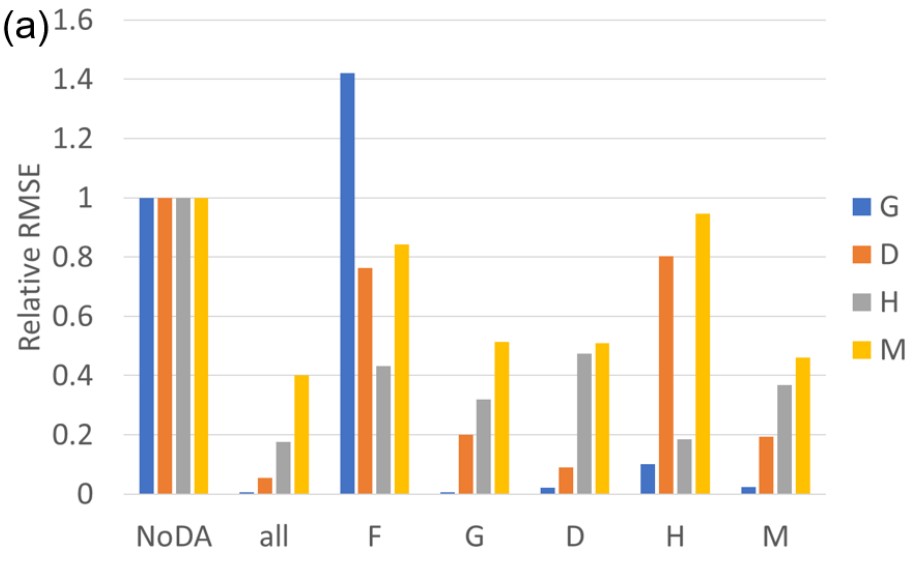

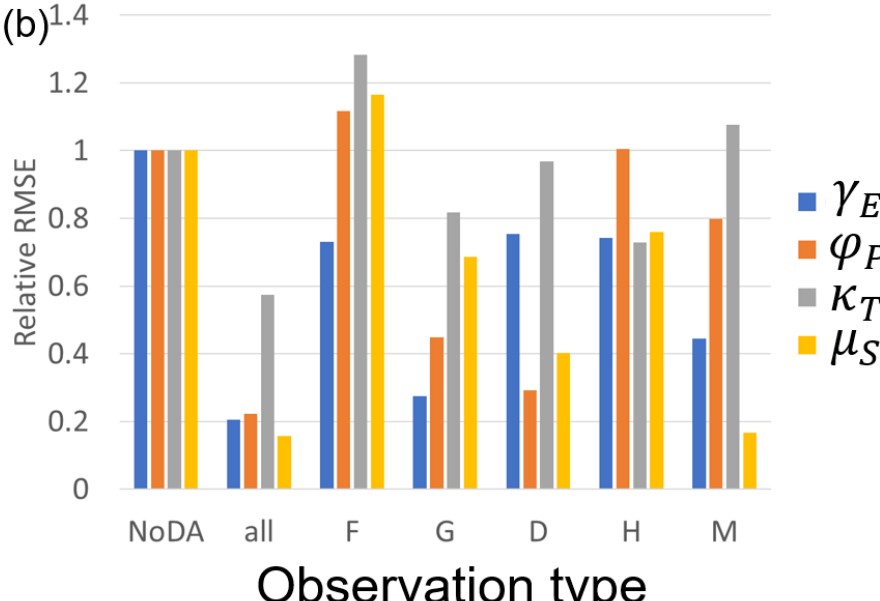


**Figure 8.** The ratio of RMSEs of the no data assimilation experiment (NoDA) to those of the data assimilation
experiments in which all of observations (F, G, D, H, and M) are assimilated (all) and each one of them is
assimilated in the experiment 2 (see section 3.2). (a) Blue, orange, gray, and yellow bars are RMSEs of the



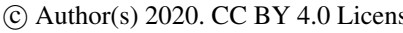



size of the human settlement G(t), the center of mass of the human settlement from the river D(t), the flood
protection level (or levee height) H(t), and the social awareness of the flood risk M(t). (b) Blue, orange, gray,
and yellow bars are RMSEs of the cost of levee raising $\gamma_E$, the rate by which new properties can be built $\varphi_P$,
the rate of decay of levees $\kappa_T$, memory loss rate $\mu_S$.




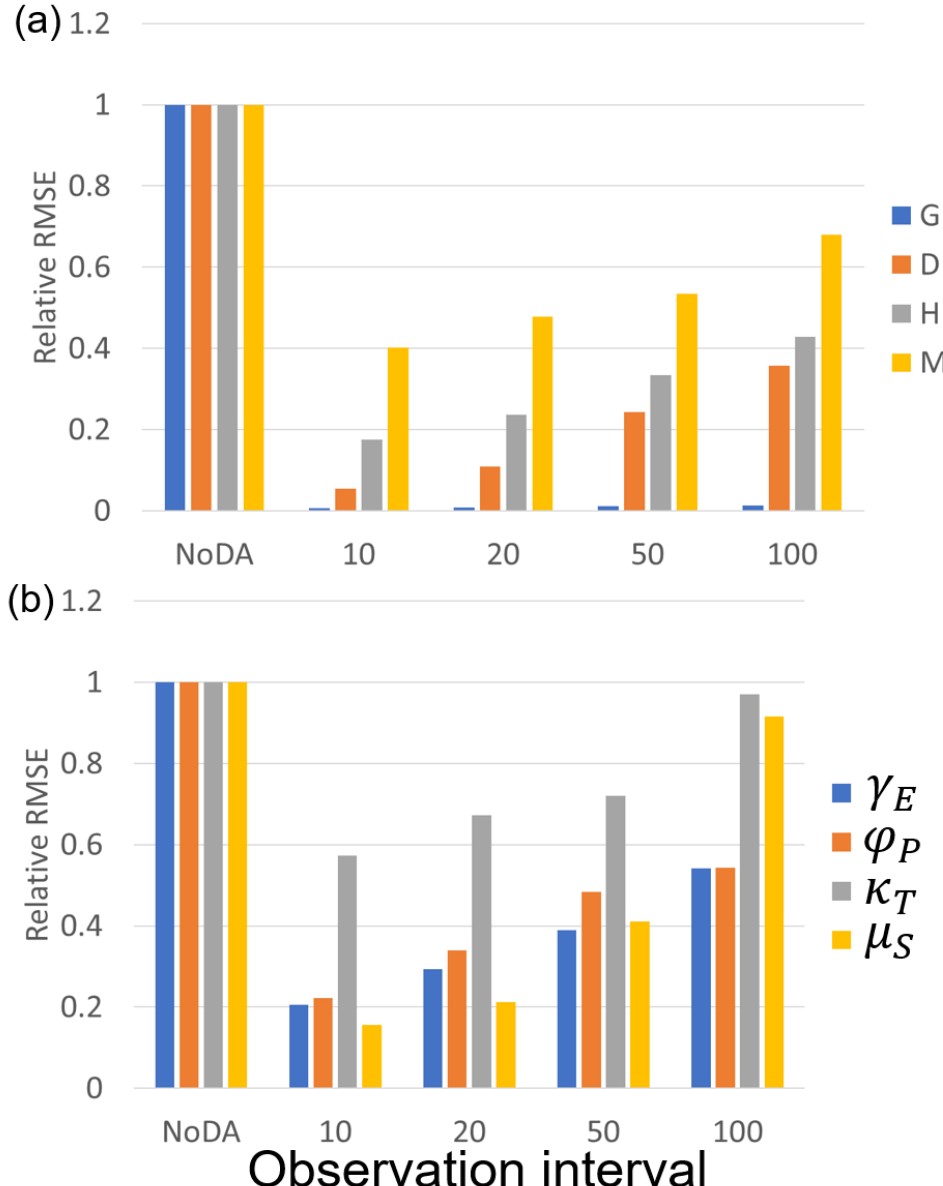


**Figure 9.** The ratio of RMSEs of the no data assimilation experiment (NoDA) to those of the data assimilation

experiments in which all of observations (F, G, D, H, and M) are assimilated every 10, 20, 50, and 100 years

in the experiment 2 (see section 3.2). (a) Blue, orange, gray, and yellow bars are RMSEs of the size of the





human settlement G(t), the center of mass of the human settlement from the river D(t), the flood protection
level (or levee height) H(t), and the social awareness of the flood risk M(t). (b) Blue, orange, gray, and yellow
bars are RMSEs of the cost of levee raising $\gamma_E$, the rate by which new properties can be built $\varphi_P$, the rate of
decay of levees $\kappa_T$, memory loss rate $\mu_S$.







**Figure 10.** Timeseries of (a) high water level W(t), (b) the flood protection level (or levee height) H(t), (c) the

distance of the center of mass of the human settlement from the river D(t), (d) the size of the human settlement

G(t), (e) the intensity of flooding events F(t), and (f) the social awareness of the flood risk M(t) simulated by

the data assimilation experiment in which the observations of F, G, D, H, and M are assimilated into the model



every 10 years with 5000 ensembles in the experiment 3 (see section 3.3).Grey, red, and black lines are the

ensemble members, their mean, and the synthetic truth, respectively.





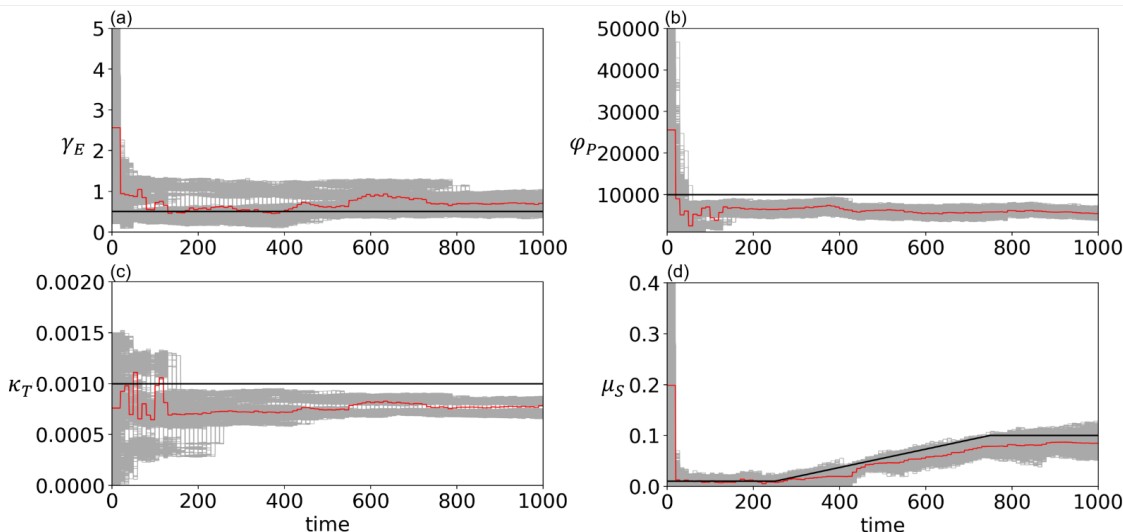

**Figure 11.** Timeseries of (a) the cost of levee raising $\gamma_E$, (b) the rate by which new properties can be built

$\varphi_P$, (c) the rate of decay of levees $\kappa_T$, (d) memory loss rate $\mu_S$ estimated by the data assimilation of all

observations (F, G, D, H, and M) with 5000 ensembles every 10 years in the experiment 3 (see section 3.3).

Grey, red, and black lines are the ensemble members, their mean, and the synthetic truth, respectively.