# Peer review of "Socio-hydrologic data assimilation: Analyzing human-flood interactions by model- data integration"

_Hydrology and Earth System Sciences, 2020_

## Referee Comment (RC1) · Anonymous Referee #1 · 21 Mar 2020

The paper "Socio-hydrologic data assimilation: Analyzing human-flood interactions by model data integration" applied a sequential data assimilation approach to the socio-hydrological model developed by Di Baldassarre et al. (2013) to update model state and estimate the model parameters. While I found the idea of combining data assimilation with socio-hydrological modeling interesting, I believe that there are different shortcomings that prevent the publication of the paper in the present form. I have provided more comments below to help the authors strengthening their paper:

- My main concern is related to the use of synthetic observations to test the hindcast assimilation experiment. On the one hand, it is a standard procedure to use synthetic

experiments to test new approaches and observations. On the other hand, synthetic experiments must be coupled with real-world analysis when observations are available. The authors state: "Although our experiment design was idealized, this study reveals several important findings toward real-world applications" (in the Discussion section). However, authors do not provide any empirical comparison or validation of their approach with real-world applications. Does the model perform better against any real-world criteria? The main reason for integrating observations into a mathematical model is to improve the representation of reality by updating model states, output, parameters or input. If this new modeling approach cannot be applied to a single case study then, what is it the purpose of this study if not just a mere numerical exercise? For example, you could use the same data reported in Barendrecht et al. (2019) to test your assimilation approach.

- The proper estimation of the observational error plays a key role in the assimilation performances (especially in socio-hydrological models in which uncertainty of social observation can be quite high). How is the covariance matrix of the observation error process estimated in this study? Are some observations more reliable than others? Small observational errors can force the updated model closer to the observation, while high observational errors may lead to poor model updates. Are the good results achieved in the experiments due to low observational error with respect to the model error with NoDA? The authors mentioned that M is considered as an observed variable for assimilation purposes. How are the authors planning to estimate the accuracy of flood awareness observations in real-world applications?

- I have some doubts about the setup of the experiments. Why only 4 parameters and 1 parameter are considered in the second and third experiments respectively? Why the authors selected those parameters and not others? This must be explained as the results can be biased by the selection of the parameters. Personally, the way I would structure the experiments (and results) of this study is 1) Uncertain input and uncertain observation (assuming different observation errors); 2) Temporal uneven distribution of

observational data (similar to Figure 4); 3) Assimilation strategies (similar to Figure 3); 4) Real-world application. I have already explained the reason behind points 1 and 4. I have included more comments on point 2 below.

- It is mentioned that "our SIRPF can efficiently improve the simulation of the socio-hydrologic state variables using the sparsely distributed data". I guess the authors refer to the temporal availability of observation and not spatial as the system dynamic model of Di Baldassarre et al. (2013) is lumped. I appreciate that the authors considered the effect of different assimilation updating times. However, I would find more interesting to consider intermittent and uneven temporal distributions of social and hydraulic information. In fact, it can be that flood awareness and other social data are not available regularly available as you assumed in your study. How would the intermittency characteristics of social data affect the model performances?

- Results need to be discussed in a more critical way. For example, why when only G is updated also the other state variables are improved? Also "Observing F, D, and M negatively impacts the estimation of H and observing H does not significantly improve the simulation of D and M". I found these results counter-intuitive. If a flood occurs, then flood awareness will increase and this will lead to reinforcement of the levee system (as already described in Di Baldassarre et al. (2013), Di Baldassarre et al. (2015), and other related papers). However, you found a negative impact. Why? Levees are built and reinforced to protect urbanized areas from flooding and F should be critical for the estimation of H. In the same way, levee systems (H) can shape human flood awareness and distance from the river (as already described in Di Baldassarre et al. (2013), Di Baldassarre et al. (2015), and many other related papers). So, how can you justify your results? Provide a real-world example in which flooding and flood awareness are not relevant to the reinforcement of a levee system.

- Line 268: "In the first OSSE, we assumed that the model was perfect, and we knew it". I honestly doubt that the model is perfect and I strongly invite the authors to remove this sentence as conceptually wrong. No model is perfect, in particular socio-hydrological

models that represent complex social interactions as the distance from the river, flood awareness, etc.

- What do you mean with "H is decoupled from the other state variables"?

- Line 298-300. The sentence is unclear and has to be rephrased.

- Line 270: "Our SIRPF estimated only state variables". Change "estimated" with "updated".

---

## Referee Comment (RC2) · Anonymous Referee #2 · 24 Mar 2020

This paper presents a study of data assimilation based on a conceptual socio-hydrologic model. The authors used the SIRPF method to assimilate human-flood interaction data based on the flood risk model developed by Di Baldassarre et al. (2013). The manuscript is well-written and the study topic is of interest to the audience of HESS. I have the following comments that I hope the authors could address in their revision. Specific comments:

1. Lines 251-252: The authors should be clear about the time scale of the model, which I assume is annual. The human-flood interactions will be different at different time scales. Also, in the time series figures, the authors should make clear statement

about the annual time step.

2. In the Results section, the authors provided interpretations of the experiment results. It would be helpful if the study can include some validation of the method. For example, the authors could apply their proposed method in a realistic case study.

3. Section 4.3, the discussion about the experiment 3 results is too general. The study could include more temporally changing variables in experiment 3 (the cost of levee raising, the rate of new properties, and the decay rate of levees), since they are all changing with time in reality.

―――――――――――――――――――――

---

## Author Comment (AC1) · 30 Mar 2020

**Real-world experiment**

**Experiment design**

In addition to the OSSEs, we performed the real-world experiment in the city of Rome,

Italy. Ciullo et al. (2017) collected real-world data and calibrated their flood risk model.

Using the data collected by Ciullo et al. (2017), we performed the data assimilation experiment. It should be noted that the flood risk model of Ciullo et al. (2017) is different from our model (i.e. Di Baldassarre et al. 2013), although they are conceptually similar.

All the data were collected from Figure 1 of Ciullo et al. (2017) by WebPlotDigitizer (https://automeris.io/WebPlotDigitizer/). The observed high water level of Tiber River was used as input forcing data (W). The levee height (H) and population (G) were used as the observation data to be assimilated into the flood risk model. In Ciullo et al. (2017), population values within the Tiber's floodplain were normalized by the theoretical maximum Tiber's floodplain population which is estimated to the range between $10^6$

and $2 \times 10^6$. Since our flood risk model needs the population values (not normalized values), we multiplied $1.5 \times 10^6$ and the normalized values shown in Figure 1 of Ciullo et al. (2017) to obtain population in the floodplain.

We added lognormal multiplicative noise to the observed high water level as we did in the OSSEs. The observation errors of levee height and population were set to 10% and

25% of the observed values, respectively. Since Ciullo et al. (2017) showed the large uncertainty in the estimation of the theoretical maximum population (see above), it is reasonable to assume that the estimation of population values also has relatively large uncertainty.

As the second and third OSSEs, we have 4 unknown parameters in this real-world experiment. We used the same settings of parameters as the OSSEs, which are shown in

[revised manuscript text omitted]

$\varphi_P$, (c) the rate of decay of levees $\kappa_T$, (d) memory loss rate $\mu_S$ estimated by the data assimilation of observations of G and H with 5000 ensembles in the real-world experiment in the city of Rome. Grey and red lines are the ensemble members and their mean, respectively.

[Figure]

**Figure S2**. Same as Figure S1 but only real data of G are assimilated.

[Figure]

**Figure S3**. Same as Figure S1 but only real data of H are assimilated.

---

## Author Comment (AC2) · 25 Apr 2020

Response letter of hess-2020-19-RC1

Dear Anonymous Referee #1,

Please find the responses to the comments.

Comments made by the reviewer were highly insightful. They allowed us to greatly improve the quality of the manuscript. We described the response to the comments.

Each comment made by the reviewers is written in *italic* font. We numbered each comment as (n.m) in which n is the reviewer number and m is the comment number. In the revised manuscript, changes are highlighted in yellow.

We trust that the revisions and responses are sufficient for our manuscript to be published in *Hydrology and Earth System Sciences*

**Responses to the comments of Referee #1**

*The paper "Socio-hydrologic data assimilation: Analyzing human-flood interactions by model data integration" applied a sequential data assimilation approach to the sociohydrological model developed by Di Baldassarre et al. (2013) to update model state and estimate the model parameters. While I found the idea of combining data assimilation with socio-hydrological modeling interesting, I believe that there are different shortcomings that prevent the publication of the paper in the present form. I have provided more comments below to help the authors strengthening their paper:*

*(1.1) My main concern is related to the use of synthetic observations to test the hindcast assimilation experiment. On the one hand, it is a standard procedure to use synthetic experiments to test new approaches and observations. On the other hand, synthetic experiments must be coupled with real-world analysis when observations are available. The authors state: "Although our experiment design was idealized, this study reveals several important findings toward real-world applications" (in the Discussion section). However, authors do not provide any empirical comparison or validation of their approach with real-world applications. Does the model perform better against any real-world criteria? The main reason for integrating observations into a mathematical model is to improve the representation of reality by updating model states, output, parameters or input. If this new modeling approach cannot be applied to a single case study then, what is it the purpose of this study if not just a mere numerical exercise? For example, you could use the same data reported in Barendrecht et al. (2019) to test your assimilation approach.*

→ Thank you very much for this comment. We have performed the real-data experiment using the data collected by Ciullo et al. (2017). The results have already been shown in the other Authors' comment. We attached them below as the proposal of the revision. We successfully showed that our SIRPF can be applied to the real-world case, which we believe significantly strengthen the paper.

[revised manuscript text omitted]

*(1.2) The proper estimation of the observational error plays a key role in the assimilation performances (especially in socio-hydrological models in which uncertainty of social observation can be quite high). How is the covariance matrix of the observation error process estimated in this study? Are some observations more reliable than others? Small observational errors can force the updated model closer to the observation, while high observational errors may lead to poor model updates. Are the good results achieved in the experiments due to low observational error with respect to the model error with NoDA? The authors mentioned that M is considered as an observed variable for assimilation purposes. How are the authors planning to estimate the accuracy of flood awareness observations in real-world applications?*

→ We assumed that the observation errors (standard deviation) were set to 10% of the true value. We mentioned this point in the original version of the paper.

"The variance of the Gaussian white noise was 10% of the synthetic true variables."

We noticed that this description was wrong. We should say "standard deviation" (not variance) here. In addition, we realized that this description was unclear. We have modified this point as follows.

"The observation error, the standard deviation of the Gaussian white noise, was firstly set to 10% of the synthetic true variables."

We believe that 10% observation error is much larger than the observation error generally used in the other earth science domains such as atmospheric science and hydrology. As the referee mentioned, we assumed that the uncertainty of social observations is quite high. This point was indeed unclear in the original version of the paper and we have clarified it in the revised version of the paper.

"Although this observation error is generally larger than that used in meteorology and hydrology,"

In addition, we tested the sensitivity of the observation error to the SIRPF's performance and found that our SIRPF is robust to the uncertain observation. In the revised version of the paper, we included

Figures S2 and S6 in the supplement material and explained our SIRPF significantly improves the state and parameter estimation with larger observation errors. As the referee mentioned, the SIRPF's performance gradually declines as the observation error increases. We have included the following sentences in the results section of the revised paper.

"Although this observation error is generally larger than that used in meteorology and hydrology, we further increased the observation error and tested the sensitivity of the observation error to the SIRPF's performance."

"We set the observation error to 10% of the synthetic truth thus far. The improvement of the simulation skill can be found with larger observation errors (Figure S2). Although the SIRPF's performance gradually declines as the observation error increases, our SIRPF can significantly improve the simulation skill with 25% observation error."

"As we found in the experiment 1, the SIRPF's performance declines with the increased observation error (Figure S6). However, it is promising that our SIRPF can improve the simulation skill with larger observation errors up to 25% of the synthetic truth considering that the observations in the socio-hydrologic domain are often inaccurate."

[Figure]

**Figure S2**. The ratio of RMSEs of the no data assimilation experiment (NoDA) to those of the data assimilation experiments in which the observation error is set to 10%, 15%, 20%, and 25% of the synthetic true values in the experiment 1 (see section 3.1.1). Blue, orange, gray, and yellow bars are RMSEs of the size of the human settlement G(t), the center of mass of the human settlement from the river D(t), the flood protection level (or levee height) H(t), and the social

[Figure]

**Figure S6.** The ratio of RMSEs of the no data assimilation experiment (NoDA) to those of the data assimilation experiments in which the observation error is set to 10%, 15%, 20%, and 25% of the synthetic true values in the experiment 2 (see section 3.1.2). (a) Blue, orange, gray, and yellow bars are RMSEs of the size of the human settlement G(t), the center of mass of the human settlement from the river D(t), the flood protection level (or levee height) H(t), and the social awareness of the flood risk M(t). (b) Blue, orange, gray, and yellow bars are RMSEs of the cost of levee raising $\gamma_E$, the rate by which new properties can be built $\varphi_P$, the rate of decay of levees $\kappa_T$, memory loss rate $\mu_S$.

We agree with the referee's comment that it is not straightforward to observe flood awareness. Several studies have obtained the proxy of the social memory by interview data (Barendrecht et al. 2019) and the number of Google searches (Gonzales and Ajami 2017). This point was indeed unclear in the original version of the paper and we have clarified this point in the revised version of the paper.

> "Although it is not straightforward to observe social memory M, several previous studies obtained the proxy of the social memory by interview data (Barendrecht et al. 2019) and the number of Google searches (Gonzales and Ajami 2017)."

*(1.3) I have some doubts about the setup of the experiments. Why only 4 parameters and 1 parameter are considered in the second and third experiments respectively? Why the authors selected those parameters and not others? This must be explained as the results can be biased by the selection of the parameters. Personally, the way I would structure the experiments (and results) of this study is 1) Uncertain input and uncertain observation (assuming different observation errors); 2) Temporal uneven distribution of observational data (similar to Figure 4); 3) Assimilation strategies (similar to Figure 3); 4) Real-world application. I have already explained the reason behind points 1 and 4. I have included more comments on point 2 below.*

→ The referee mentioned that only 1 parameter were considered in the third experiments. Please note that we actually considered all 4 parameters and one of the 4 parameters were assumed to be time-variant. As the response to the comment from the referee #2, we will include one more parameter as time-variant parameters in the revised version of the paper, if we are allowed to revise. Please see our response to the comment (2.3) of the referee #2.

We believe that the selection of the targeted parameters in socio-hydrologic data assimilation will depend on the case and purpose of the study. The problem setting adopted in this study can be recognized as one of the reasonable examples without the significant loss of generality. Here we explain how to select those parameters as a reasonable example of socio-hydrologic data assimilation. First, it is unlikely that the parameters related to F in equation (1) are much more inaccurate than the other parameters. They are mainly determined by the topography and we believe the process described in equation (1) can be replaced by the more accurate hydrodynamic models. Second, we selected four unknown parameters one by one from four equations of economy, politics, technology, and social to discuss how each state variable's observation affects the parameter space. Third, our 4 unknown parameters, their initial uncertainties, and the uncertainty in the high water level make our problem difficult enough to demonstrate the potential of data assimilation in the socio-hydrologic domain.

Figure 5 indicates that we can get no useful information of the socio-hydrologic processes with this specified uncertainty. Although the referee may think that the number of unknown parameters is too small, we believe that our problem gives enough uncertainty to demonstrate the potential of data assimilation. Fourth, we successfully applied this setting to the real-world case in the city of Rome so that our specified initial uncertainty is reasonably good. We have added some sentences to explain this point in the revised version of the paper.

"We selected these unknown parameters one by one from four equations of economy, politics, technology, and social to discuss how each state variable's observation affects the estimation of parameters across these four equations (see section 2.1). We have no unknown parameters related to F (equation (1)) since it is unlikely that the parameters in equation (1) are much more inaccurate than the other parameters. The parameters related to flood are mainly determined by the topography of the flood plain so that the process described in equation (1) can be replaced by more accurate hydrodynamic models in the real-world case study. The initial parameter variables were assumed to be distributed in the bounded uniform distributions whose ranges were found in Table 1. The uncertainty of the simulation induced by these parameters' uncertainty is large enough to demonstrate the potential of data assimilation to minimize the simulation's uncertainty (see Results)."

The referee suggested to changing the structure of the paper in the latter part of this comment. We would like to keep the structure of the original paper because in the current structure, the problem setting gets harder and approaches to the real-world problem (and eventually arrive at the real-data experiment). We believe that the referee's concerns have been addressed by out responses to the comments. Please see our responses to the comments (1.1) (real-data experiment), (1.2) (observation error) and (1.4) (temporally uneven observation). We believe the change in the structure of the paper is not absolutely necessary to meet the referee's requirements. We have decided not to change this aspect of the paper.

*(1.4) It is mentioned that "our SIRPF can efficiently improve the simulation of the sociohydrologic state variables using the sparsely distributed data". I guess the authors refer to the temporal availability of observation and not spatial as the system dynamic model of Di Baldassarre et al. (2013) is lumped. I appreciate that the authors considered the effect of different assimilation updating times. However, I would find more interesting to consider intermittent and uneven temporal distributions of social and hydraulic information. In fact, it can be that flood awareness and other social data are not available regularly available as you assumed in your study. How would the intermittency characteristics of social data affect the model performances?*

→ I believe that our new real-data experiment shows the reasonable performance of our SIRPF with the intermittent and uneven temporally distributed observation. Since we need no tuning of the hyperparameters with different observation intervals in our SIRPF, we have no problem when the observation intervals temporally change. As Figures S1 and S3 indicate, the uncertainty (i.e. the spread of ensembles) increases in the observation interval so that we have relatively large (small) uncertainty if we have a large (small) observation interval. This finding is consistent to the unevenly distributed observation in the real-data experiment. We believe that the real-data experiment has already addressed this comment and the additional OSSE is not necessary. It is not straightforward to design the realistic unevenly distributed observation in the framework of OSSE so that it is better to show the real-data experiments to explain our SIRPF is robust to this issue. In the revised version of the paper, we emphasized this point as follows.

> "In contrast to the OSSEs, our observation network has the uneven temporal distribution. Figure 13 clearly indicates that our SIRPF is robust to these intermittent observations whose intervals temporally change."

*(1.5) Results need to be discussed in a more critical way. For example, why when only G is updated also the other state variables are improved? Also "Observing F, D, and M negatively impacts the estimation of H and observing H does not significantly improve the simulation of D and M". I found these results counter-intuitive. If a flood occurs, then flood awareness will increase and this will lead to reinforcement of the levee system (as already described in Di Baldassarre et al. (2013), Di Baldassarre et al. (2015), and other related papers). However, you found a negative impact. Why? Levees are built and reinforced to protect urbanized areas from flooding and F should be critical for the estimation of H. In the same way, levee systems (H) can shape human flood awareness and distance from the river (as already described in Di Baldassarre et al. (2013), Di Baldassarre et al. (2015), and many other related papers). So, how can you justify your results? Provide a real-world example in which flooding and flood awareness are not relevant to the reinforcement of a levee system.*

→ The first question of this block is why the other state variables are improved by updating only G. The referee has a misunderstanding due to our insufficient description. We actually updated all state variables (and parameters) by assimilating only G. Although we evaluated equation (11) using only simulated and observed G, x(t) in (13) includes all state variables (and parameters). In SIRPF, we can infer the unobserved variables from the observed variables and simulation based on the Bayes' theorem. This point was indeed unclear in the original version of the paper. We have clarified this point in the revised version of the paper.

"Note that equations (13) and (14) update all state variables and parameters of the model although the weight is calculated using only observable variables. Therefore, it is not necessary to observe all state variables in order to update all system variables."

"Our SIRPF updates all state variables although only one of them is assimilated."

The second question of this block is why the observation H has little impact on the estimation of the state space in summary. This comment is critical, and we thank the referee for this comment. In the flood risk model of Di Baldassarre et al. (2013), flooding and flood awareness strongly control whether they raise the levee or not as the referee mentioned. However, once they determined to raise the levee responding to the flood event, the amount by which the levees are raised is fully determined by the high water level (see equation (2)). On the contrary to the previous works, we assumed the uncertainty in the high water level, which brings the uncertainty in the dynamics of H. Observing H can improve the simulation of D and M only if the uncertainty in the dynamics of H is induced by the uncertainty in D and M. Because the uncertainty in the dynamics of H is largely determined by the uncertainty in the high water level, we could not obtain the significant impact of the observation of H on the other variables. This point was indeed unclear in the original version of the paper and we have clarified this point in the revised version of the paper.

"Although the dynamics of F, D, and M strongly affects the decision making of whether the levees are raised or not, the amount by which the levees are raised, R, is fully determined by the high water level, W, once the community determines to raise the levees (see equation (2)). Therefore, the uncertainty of H is largely induced by the uncertainty of the high water level, W, whose uncertainty is not directly mitigated by our SIRPF. This is why observing F, D, and M is not helpful to mitigate the uncertainty of H."

*(1.6) Line 268: "In the first OSSE, we assumed that the model was perfect, and we knew it". I honestly doubt that the model is perfect and I strongly invite the authors to remove this sentence as conceptually wrong. No model is perfect, in particular socio-hydrological models that represent complex social interactions as the distance from the river, flood awareness, etc.*

→ This is indeed the assumption and we do not trust it is true in the real-world. We understand that no model is perfect in the real-world application. In the revised version of the paper, we rephrased this sentence as follows.

"we assumed that there is no uncertainty in model parameters."

*(1.7) What do you mean with "H is decoupled from the other state variables"?*

→ We intended to make this sentence the summary (or rephrasing) of the next sentence "Observing F,

D, and M negatively impacts the estimation of H and observing H does not significantly improve the simulation of D and M.". We noticed that this sentence is simply unnecessary to explain our results. We have deleted this sentence in the revised version of the paper.

*(1.8) Line 298-300. The sentence is unclear and has to be rephrased.*

→ In the revised paper, we have rephrased this sentence in the following:

"In the data assimilation experiment, we assumed that the dynamics of $\varphi_P$ and $\mu_S$ was unknown, and we integrated the flood risk model with time-invariant $\varphi_P$ and $\mu_S$."

*(1.9) Line 270: "Our SIRPF estimated only state variables". Change "estimated" with "updated".*

→ We have modified this point following the reviewer's instruction.

---

## Author Comment (AC3) · 25 Apr 2020

Response letter of hess-2020-19-RC2

Dear Anonymous Referee #2,

Please find the responses to the comments.

Comments made by the reviewer were highly insightful. They allowed us to greatly improve the quality of the manuscript. We described the response to the comments.

Each comment made by the reviewers is written in *italic* font. We numbered each comment as (n.m) in which n is the reviewer number and m is the comment number. In the revised manuscript, changes are highlighted in yellow.

We trust that the revisions and responses are sufficient for our manuscript to be published in *Hydrology and Earth System Sciences*

**Responses to the comments of Referee #2**

*This paper presents a study of data assimilation based on a conceptual sociohydrologic model. The authors used the SIRPF method to assimilate human-flood interaction data based on the flood risk model developed by Di Baldassarre et al. (2013). The manuscript is well-written and the study topic is of interest to the audience of HESS. I have the following comments that I hope the authors could address in their revision. Specific comments:*

*(2.1) Lines 251-252: The authors should be clear about the time scale of the model, which I assume is annual. The human-flood interactions will be different at different time scales. Also, in the time series figures, the authors should make clear statement about the annual time step.*

→ This point was indeed unclear in the original version of the paper. We chose the annual time step. We have clarified this point in the model section of the revised paper.

"==The timestep was set to annual.=="

This point has also been clarified in the caption of figures.

*(2.2) In the Results section, the authors provided interpretations of the experiment results. It would be helpful if the study can include some validation of the method. For example, the authors could apply their proposed method in a realistic case study.*

→ Thank you very much for this comment. We performed the real-data experiment using the data collected by Ciullo et al. (2017). The results have already been shown in the other Authors' comment. We have also attached it below as the proposal of the revision.

[revised manuscript text omitted]

*(2.3) Section 4.3, the discussion about the experiment 3 results is too general. The study could include more temporally changing variables in experiment 3 (the cost of levee raising, the rate of new properties, and the decay rate of levees), since they are all changing with time in reality.*

→ Thank you for this comment. We could include more temporally changing parameters as the referee indicated. In the revised version of the paper, we have included the rate by which new properties can be built, $\varphi_P$, as a time-variant parameter. We have modified the manuscript as follows:

"Specifically, the rate by which new properties can be built, $\varphi_P$, and the memory loss rate, $\mu_S$, were temporally varied in the experiment 3:

$$\varphi_P(t) = \begin{cases} 5000 \ (t < 250) \\ 5000 + (t - 250) \times \frac{40000 - 5000}{500} \ (250 \leq t < 750) \\ 40000 \ (750 \leq t) \end{cases} \quad (24)$$

$$\mu_S(t) = \begin{cases} 0.01 \ (t < 250) \\ 0.01 + (t - 250) \times \frac{0.10 - 0.01}{500} \ (250 \leq t < 750) \\ 0.10 \ (750 \leq t) \end{cases} \quad (25)$$

In the data assimilation experiment, we assumed that the dynamics of $\varphi_P$ and $\mu_S$ was unknown, and we integrated the flood risk model with time-invariant $\varphi_P$ and $\mu_S$."

"In addition to the experiment 2, two of the unknown parameters ($\varphi_P$ and $\mu_S$) temporally vary in the synthetic truth of the experiment 3. We found that a larger spread of $\varphi_P$ is required to stably track the time-variant synthetic true $\varphi_P$ so that we increased $s_0$ in equation (18) from 0.05 to 0.5 only for $\varphi_P$ in this experiment 3. Figure 10 and Table 4 indicate that despite the error in the model's description, our SIRPF can greatly improve the simulation of the flood risk model.

Please note that the synthetic truth shown in Figure 10 is different from that of the previous experiments especially for D and M. Figures 11b and 11d indicate that we can accurately estimate the time-variant parameters ($\varphi_P$ and $\mu_S$) as well as the other time-invariant parameters (Figures 11a and 11c). This result is promising since we cannot expect the perfect description of the socio-hydrologic model in the real-world applications. We also performed the sensitivity test on observation types, observation intervals, and ensemble sizes, which results in the same conclusions as the experiment 2 (not shown)."

"

[Figure]

**Figure 10.** Timeseries of (a) high water level W(t), (b) the flood protection level (or levee height) H(t), (c) the distance of the center of mass of the human settlement from the river D(t), (d) the

size of the human settlement G(t), (e) the intensity of flooding events F(t), and (f) the social awareness of the flood risk M(t) simulated by the data assimilation experiment in which the observations of F, G, D, H, and M are assimilated into the model every 10 years with 5000 ensembles in the experiment 3 (see section 3.1.3). The time step is annual. Grey, red, and black lines are the ensemble members, their mean, and the synthetic truth, respectively.

[Figure]

**Figure 11.** Timeseries of (a) the cost of levee raising $\gamma_E$, (b) the rate by which new properties can be built $\varphi_P$, (c) the rate of decay of levees $\kappa_T$, (d) memory loss rate $\mu_S$ estimated by the data assimilation of all observations (F, G, D, H, and M) with 5000 ensembles every 10 years in the experiment 3 (see section 3.1.3). The time step is annual. Grey, red, and black lines are the ensemble members, their mean, and the synthetic truth, respectively."

The other two parameters, the cost of levee raising and the decay rate of levees, were still kept constant in the synthetic truth. This is because the temporal change in these two parameters has small impacts on the state variables, which make it difficult to sequentially estimate the temporal change of the parameters. The cost of levee raising mainly affects the state variables in the early stage of the simulation and the change in the decay rate of levees has much smaller impacts than the uncertainty of high water level. This point was indeed unclear in the original version of the paper although it should be mentioned as the limitation. In the revised version of the paper, we have included this point as follows:

"The cost of levee raising $\gamma_E$ affects the state variables of the flood risk model mainly in the initial early years and the gradual change of the rate of decay of levees $\kappa_T$ has few impacts on

the state variables. Therefore, we found that it is difficult to track the temporal change of these two parameters."

As we discuss in the response to the comment from the referee #1 (comment (1.3)), the problem setting of the parameter estimation strongly depends on the case and purpose of the study. We believe that our current problem setting is one of the reasonable examples without the significant loss of generality. The referee mentioned that all parameters may be time-variant. Although we can agree with it, if all parameters of the model can be time-variant, the model provides no reasonable constraints of the trajectory of the state variables. In this case, we doubt that it is beneficial to use the model to analyze the socio-hydrologic phenomena. Therefore, we believe that it is reasonable to assume some of the parameters are known and only a few parameters are time-variant. Please also see the discussion with the referee #1 attached below.
* * *
*(1.3) I have some doubts about the setup of the experiments. Why only 4 parameters and 1 parameter are considered in the second and third experiments respectively? Why the authors selected those parameters and not others? This must be explained as the results can be biased by the selection of the parameters. Personally, the way I would structure the experiments (and results) of this study is 1) Uncertain input and uncertain observation (assuming different observation errors); 2) Temporal uneven distribution of observational data (similar to Figure 4); 3) Assimilation strategies (similar to Figure 3); 4) Real-world application. I have already explained the reason behind points 1 and 4. I have included more comments on point 2 below.*

→ The referee mentioned that only 1 parameter were considered in the third experiments. Please note that we actually considered all 4 parameters and one of the 4 parameters were assumed to be time-variant. As the response to the comment from the referee #2, we will include one more parameter as time-variant parameters in the revised version of the paper, if we are allowed to revise. Please see our response to the comment (2.3) of the referee #2.

We believe that the selection of the targeted parameters in socio-hydrologic data assimilation will depend on the case and purpose of the study. The problem setting adopted in this study can be recognized as one of the reasonable examples without the significant loss of generality. Here we explain how to select those parameters as a reasonable example of socio-hydrologic data assimilation. First, it is unlikely that the parameters related to F in equation (1) are much more inaccurate than the other parameters. They are mainly determined by the topography and we believe the process described in equation (1) can be replaced by the more accurate hydrodynamic models. Second, we selected four

unknown parameters one by one from four equations of economy, politics, technology, and social to discuss how each state variable's observation affects the parameter space. Third, our 4 unknown parameters, their initial uncertainties, and the uncertainty in the high water level make our problem difficult enough to demonstrate the potential of data assimilation in the socio-hydrologic domain. Figure 5 indicates that we can get no useful information of the socio-hydrologic processes with this specified uncertainty. Although the referee may think that the number of unknown parameters is too small, we believe that our problem gives enough uncertainty to demonstrate the potential of data assimilation. Fourth, we successfully applied this setting to the real-world case in the city of Rome so that our specified initial uncertainty is reasonably good. We have added some sentences to explain this point in the revised version of the paper.

> "We selected these unknown parameters one by one from four equations of economy, politics, technology, and social to discuss how each state variable's observation affects the estimation of parameters across these four equations (see section 2.1). We have no unknown parameters related to F (equation (1)) since it is unlikely that the parameters in equation (1) are much more inaccurate than the other parameters. The parameters related to flood are mainly determined by the topography of the flood plain so that the process described in equation (1) can be replaced by more accurate hydrodynamic models in the real-world case study. The initial parameter variables were assumed to be distributed in the bounded uniform distributions whose ranges were found in Table 1. The uncertainty of the simulation induced by these parameters' uncertainty is large enough to demonstrate the potential of data assimilation to minimize the simulation's uncertainty (see Results)."

The referee suggested to changing the structure of the paper in the latter part of this comment. We would like to keep the structure of the original paper because in the current structure, the problem setting gets harder and approaches to the real-world problem (and eventually arrive at the real-data experiment). We believe that the referee's concerns have been addressed by out responses to the comments. Please see our responses to the comments (1.1) (real-data experiment), (1.2) (observation error) and (1.4) (temporally uneven observation). We believe the change in the structure of the paper is not absolutely necessary to meet the referee's requirements. We have decided not to change this aspect of the paper.

---

## Author Response (AR2)

Response letter of hess-2020-19-Report #2

Dear Anonymous Referee #1,

Please find the responses to the comments.

Comments made by the reviewer were highly insightful. They allowed us to greatly improve the quality of the manuscript. We described the response to the comments.

Each comment made by the reviewers is written in *italic* font. We numbered each comment as (n.m) in which n is the reviewer number and m is the comment number. In the revised manuscript, changes are highlighted in yellow.

We trust that the revisions and responses are sufficient for our manuscript to be published in *Hydrology and Earth System Sciences*

**Responses to the comments of Referee #1**

*First of all, I would like to thank the authors for having carefully addressed all my comments. I also appreciate their effort in proposing a real-world application based on the data of Ciullo et al. (2017). However, I still do have a few comments.*

*(2.1) - In my previous review, I asked to clarify how are the authors planning to estimate the accuracy of flood awareness observations. The authors replied that "several previous studies obtained the proxy of the social memory by interview data (Barendrecht et al. 2019) and the number of Google searches (Gonzales and Ajami 2017)". However, it is still not clear to me how is the authors' modeling framework going to assimilate such social observations and how are they going to assign an error to such observations. Maybe this is a limitation that should be included in the discussion of the results.*

→ We fully agree with this comment. When the modelled state variables cannot be directly observed, it is not straightforward to assimilate observations into a model. Particle filter and any other state-of-the-art data assimilation methods are generally flexible to this case since the nonlinear map h in equation (9) can deal with the complex relationship between model states and observable variables. In numerical weather prediction, it is an active research area to consider how to design the nonlinear map h and how to assign the observation error especially when we assimilate satellite observations. Using these previous findings, we should consider how to assimilate the indirect observation of social awareness as future work. This point was indeed unclear in the original version of the paper. We have clarified this point in the revised version of the paper.

> Lines 519-530: "The major limitation of this study is that we assume the modeled state variables can directly be observed although it is difficult to directly observe state variables of the socio-hydrologic models. For example, it is impossible to directly observe social awareness of flood risk in the flood risk model and several previous studies obtained the proxy of the social memory by interview data (Barendrecht et al. 2019) and the number of Google searches (Gonzales and Ajami 2017). When these indirect observations are assimilated into a model, the (non-linear) observation operator (see equation (9)), the assignment of the observation error, and assimilation methods should be carefully designed as previously discussed in the context of numerical weather prediction (e.g., Sawada et al. 2019; Okamoto et al. 2019; Minamide and Zhang 2017). Future work will focus on the methodological development to efficiently assimilate observations in the social domain with complicated structure of observation operators and errors."

*(2.2) - I think it would be really interesting to read more about the use of such assimilation framework to better understand the human-flood dynamics. Right now the discussion of the results is more focused*

*on the numerical tool, its performances, and observations availability (which is great). However, in my opinion, it would be also interesting to discuss more in detail how such a tool could help us in advancing our understanding of the complex feedback between the human and flood systems. What about using the proposed approach for predictions in socio-hydrological modeling?*

→ We fully agree with this comment. We believe that socio-hydrologic data assimilation is useful to reconstruct the historical human-flood interactions which includes unobservable state variables. In the atmospheric science, atmospheric reanalysis has been intensively analyzed to understand complex feedback between many physical processes in the atmosphere, which cannot be done by simply analyzing observation data due to their sparsity. As we do with the atmospheric reanalysis, socio-hydrologic reanalysis works as a reliable and spatio-temporally homogeneous dataset and may be helpful to deepen our understanding of human and flood. In addition, as we do with the atmospheric reanalysis, we can use the socio-hydrologic reanalysis as the initial condition to predict the future of socio-hydrologic processes. It is impossible to obtain the complete set of state variables and parameters by observation due to its sparsity so that data assimilation contributes to generating good initial conditions and future projection. Although we have already mentioned the concept of "socio-hydrologic reanalysis" in the original paper, this point was indeed unclear. We have clarified this point in the revised version of the paper.

> Lines 581-592: "In the atmospheric science, atmospheric reanalysis has been intensively analyzed to understand complex feedback in the atmosphere, which cannot be done by analyzing only observation data due to their sparsity. Socio-hydrologic reanalysis can work as a reliable and spatio-temporally homogeneous dataset and may be helpful to deepen our understanding of human and water. In addition, socio-hydrologic reanalysis can be used as initial condition to predict the future change of socio-hydrologic processes as atmospheric scientists predict the future weather/climate using atmospheric reanalysis. Since it is impossible to directly observe all state variables and parameters as initial condition, socio-hydrologic reanalysis is crucially important for accurate prediction. Socio-hydrologic data assimilation has a high potential to improve our understanding of the complex feedback between social and flood systems and predict their future."

*(2.3) - In the title use either "assimilation" or "integration", they are synonymous but they do mean different things.*

→ Data assimilation is the technical term which indicates approaches to sequentially estimate the state from observations and model based on their errors. In scientific papers, data assimilation includes the specific methods such as particle filter, ensemble Kalman filter, and 4-D variational methods. Therefore, we cannot say data "integration". On the other hand, we believe that model-data integration can be used as a broader concept that includes the methods to estimate and understand the phenomena using both model and data. Note that model-data "assimilation" has not been used in the literature. What we do in this paper is data assimilation so that it is appropriate to include data assimilation in the title. Since many scientists in socio-hydrology may not be so familiar to data assimilation, we believe that using model-data integration in the title is helpful to get the broader audiences for this paper. We do understand that they mean different things as the reviewer suggested. However, given the above, we would like to continue to use both "data assimilation" and "model-data integration" in the title. We have decided not to change this aspect of the paper.

*(2.4) - How is it possible to get awareness higher than 1?*
→ In the equation, there are no reasons why awareness should not be higher than 1. It is not a normalized variable nor a ratio so that it can be higher than 1 when its decay rate is small, and the community repeatedly experiences severe floods. Because we do not imply that M is a normalized variable in the original version of the paper, we believe that it is unnecessary to mention this point. We have decided not to change this aspect of the paper.

*(2.5) - I invite the authors to improve the overall quality of the figures and include all the information in the legend of figures 12-15 (ensemble, mean ensemble, observations).*
→ We have improved most figures with the appropriate legend.

[revised manuscript text omitted]